# Efficient Distance Approximation for Structured High-Dimensional Distributions via Learning[*]

**Arnab Bhattacharyya**
National University of Singapore
`arnabb@nus.edu.sg`

**Sutanu Gayen**
National University of Singapore
`sutanugayen@gmail.com`

**Kuldeep S. Meel**
National University of Singapore
`meel@comp.nus.edu.sg`

**N. V. Vinodchandran**
University of Nebraska-Lincoln
`vinod@cse.unl.edu`

## Abstract

We design efficient distance approximation algorithms for several classes of well-studied structured high-dimensional distributions. Specifically, we present algorithms for the following problems (where $d_{\mathrm{TV}}$ is the total variation distance):

- Given sample access to two Bayesian networks $P_1$ and $P_2$ over known directed acyclic graphs $G_1$ and $G_2$ having $n$ nodes and bounded in-degree, approximate $d_{\mathrm{TV}}(P_1, P_2)$ to within additive error $\varepsilon$ using $\mathrm{poly}(n, \varepsilon^{-1})$ samples and time.
- Given sample access to two ferromagnetic Ising models $P_1$ and $P_2$ on $n$ variables with bounded width, approximate $d_{\mathrm{TV}}(P_1, P_2)$ to within additive error $\varepsilon$ using $\mathrm{poly}(n, \varepsilon^{-1})$ samples and time.
- Given sample access to two $n$-dimensional Gaussians $P_1$ and $P_2$, approximate $d_{\mathrm{TV}}(P_1, P_2)$ to within additive error $\varepsilon$ using $\mathrm{poly}(n, \varepsilon^{-1})$ samples and time.
- Given access to observations from two causal models $P$ and $Q$ on $n$ variables that are defined over known causal graphs, approximate $d_{\mathrm{TV}}(P_a, Q_a)$ to within additive error $\varepsilon$ using $\mathrm{poly}(n, \varepsilon^{-1})$ samples and time, where $P_a$ and $Q_a$ are the interventional distributions obtained by the intervention $\mathrm{do}(A = a)$ on $P$ and $Q$ respectively for a particular variable $A$.

The distance approximation algorithms immediately imply new *tolerant closeness testers* for the corresponding classes of distributions. Prior to our work, only *non-tolerant testers* were known for both Bayes net distributions and Ising models, and no testers with quantitative guarantees were known for interventional distributions. To the best of our knowledge, efficient distance approximation algorithms for Gaussian distributions were not present in the literature. Our algorithms are designed using a conceptually simple but general framework that is applicable to a variety of scenarios.

## 1 Introduction

Machine learning is primarily concerned with the design of techniques to enable the learning of a generative model $\mathcal{M}$ given access to data $\mathcal{D}$ arising from another distribution, say $P$ [Mur12]. While $P$ is typically an unknown distribution, the design of a new ML technique is often accompanied by empirical and theoretical studies under certain assumptions on $P$. Let $Q$ be the distribution generated by $\mathcal{M}$; then ideally, one would learn $\mathcal{M}$ such $P$ and $Q$ are as close as possible. Given the widespread

---

[*]Authors are in the alphabetical order.

adoption of machine learning techniques in critical domains, there has been a surge in interest of the design of techniques for rigorous verification of machine learning systems [SSS16]. The development of such verification techniques would necessitate the development of algorithmic techniques for rigorous approximation of the distance between two distributions $P$ and $Q$.

Distance approximation is also closely related to the topic of *distribution testing* investigated in the statistics and algorithms communities. Two important testing problems are *identity testing* (or, *goodness-of-fit testing*) and *closeness testing* (or, *two-sample testing*). Given samples from an unknown distribution $P$ over a domain $\mathcal{S}$, the problem of identity testing seeks to ask whether $P$ equals a specific reference distribution $Q$. A sequence of works [Pan08, BFR$^+$13, VV17, CDVV14] in the property testing literature has pinned down the finite sample complexity of this problem. It is known that with $O(|\mathcal{S}|^{1/2}\varepsilon^{-2})$ samples from $P$, one can, with probability at least $2/3$, distinguish whether $P = Q$ or whether $d_{\mathrm{TV}}(P,Q) > \varepsilon$; also, $\Omega(|\mathcal{S}|^{1/2}\varepsilon^{-2})$ samples are necessary for this task. An important generalization of identity testing is closeness testing: given samples from two unknown distributions $P$ and $Q$ over $\mathcal{S}$, does $P = Q$? Here, $\Theta(|\mathcal{S}|^{2/3}\varepsilon^{-4/3} + |\mathcal{S}|^{1/2}\varepsilon^{-2})$ samples are necessary and sufficient to distinguish $P = Q$ from $d_{\mathrm{TV}}(P,Q) > \varepsilon$ with probability at least $2/3$. The corresponding algorithms for both identity and closeness testing run in time polynomial in $|\mathcal{S}|$ and $\varepsilon^{-1}$. However, in order to solve these testing problems in many real-life settings, there are two issues that need to be surmounted.

– **High dimensions:** In typical applications, the data is described using a huge number of (possibly redundant) features; thus, each item in the dataset is represented as a point in a high-dimensional space. If $\mathcal{S} = \Sigma^n$, then from the results quoted above, identity testing or closeness testing for arbitrary probability distributions over $\mathcal{S}$ requires $2^{\Omega(n)}$ many samples, which is clearly unrealistic. Hence, we need to restrict the class of input distributions.

– **Approximation:** A high-dimensional distribution requires a large number of parameters to be specified. So, for identity testing, it is unlikely that we can ever hypothesize a reference distribution $Q$ such that it exactly equals the data distribution $p$. Similarly, for closeness testing, two data distributions $P$ and $Q$ are most likely not exactly equal. Hence, we would like to design *tolerant* testers for identity and closeness that distinguish between the cases $d_{\mathrm{TV}}(P,Q) \leqslant \varepsilon_1$ and $d_{\mathrm{TV}}(P,Q) > \varepsilon_2$ where $\varepsilon_1 < \varepsilon_2$ are user-supplied parameters.

We address both these issues by focusing on designing distance approximation algorithms for certain classes of structured distributions over $\Sigma^n$, where $\Sigma$ is an arbitrary finite set.

**Definition 1.1.** *Let $\mathcal{D}_1, \mathcal{D}_2$ be two families of distributions over $\Sigma^n$. A distance approximation algorithm for $(\mathcal{D}_1, \mathcal{D}_2)$ is a randomized algorithm $\mathcal{A}$ which takes as input $\varepsilon \in (0,1)$, and sample access to two unknown distributions $P \in \mathcal{D}_1, Q \in \mathcal{D}_2$. The algorithm $\mathcal{A}$ returns as output a value $\gamma \in [0,1]$ such that, with probability[†] at least $2/3$:*

$$\gamma - \varepsilon \leqslant d_{\mathrm{TV}}(P,Q) \leqslant \gamma + \varepsilon.$$

*If $\mathcal{D}_1 = \mathcal{D}_2 = \mathcal{D}$, then we refer to such an algorithm as a* distance approximation algorithm for $\mathcal{D}$.

***Equivalence of distance approximation and tolerant testing:*** Designing distance approximation algorithms is essentially equivalent to designing tolerant testing algorithms. Indeed, Parnas et al. [PRR06] observed that existence of a distance approximation with sample/time complexity $F(\varepsilon, n)$ for two families of distributions implies a tolerant testing algorithm with complexity $F\left(\frac{\varepsilon_2 - \varepsilon_1}{2}, n\right)$; and conversely, existence of a tolerant testing algorithm with sample/time complexity $F(\varepsilon_2 - \varepsilon_1, n)$ implies an algorithm for distance approximation with sample/time complexity $O(\log(1/\varepsilon)\log\log(1/\varepsilon)) \cdot F(2\varepsilon, n)$. Thus, henceforth we use "distance approximation" and "tolerant testing" interchangeably.

In this work, we design the first computational and sample efficient distance approximation algorithms (equivalently tolerant testing algorithms) for a variety of structured high-dimensional distributions: Bayesian networks, Ising Models, multivariate Gaussians, and interventional distributions arising from causal Bayesian networks. Our results advance the state-of-the-art in the following way:

1. Our algorithm for testing distributions over Bayes nets extends prior work [DP17, CDKS17]. In particular, in [DP17], Daskalakis and Pan presented an algorithm for *non-tolerant* closeness

---

[†]The success probability can be amplified to $1-\delta$ by taking the median of $O(\log \delta^{-1})$ independent repetitions of the algorithm with success probability $2/3$.

testing of two Bayes net distributions $P$ and $Q$ over the same known graph [‡]. We present tolerant closeness testing algorithm for two Bayes net distributions $P$ and $Q$ over *two different graphs* that asymptotically matches the sample and time complexity of their algorithm.

2. We design efficient tolerant testers for Ising models. Our first algorithm approximates the distance between any two ferromagnetic Ising models. Our second algorithm approximates the distance between any Ising model and the uniform distribution. Previously proposed testing algorithms for Ising models by [DDK19] do not achieve non-trivial tolerance.

3. Given access to poly$(n)$ samples from two multivariate Gaussians over $\mathbb{R}^n$, it is a folklore that one can approximate the distance between them. However, that algorithm is not computationally efficient. We design the first efficient algorithm to approximate distance between two multivariate Gaussians, to the best of our knowledge.

4. Given observations from two causal models $P$ and $Q$ described by two Bayesian networks on the same variable set, we give an efficient algorithm to approximate the distance between the interventional distributions obtained by fixing a particular variable. Celebrated work of Tian and Pearl [TP02a, Tia02] gave identifiability conditions. However efficient distance approximation algorithms with finite sample guarantees were non-existent prior to our work.

All our algorithms are based on a common framework. To approximate the distance between $P \in \mathcal{D}_1$ and $Q \in \mathcal{D}_2$, we first *learn* the model parameters for $\hat{P} \in \mathcal{D}_1$ and $\hat{Q} \in \mathcal{D}_2$ that are guaranteed to be close to $P$ and $Q$ respectively. It remains to compute $d_{\mathrm{TV}}(\hat{P}, \hat{Q})$. This is a computationally hard problem in general, but we use the fact that for $\mathcal{D}_1, \mathcal{D}_2$ of interest, we can efficiently approximate the mass functions for $\hat{P}$ and $\hat{Q}$ from their parameters. At this point, we invoke an estimator that approximates $d_{\mathrm{TV}}(\hat{P}, \hat{Q})$ using samples from $P$ and the approximate mass functions for $\hat{P}$ and $\hat{Q}$.

A salient strength of our framework is its conceptual simplicity. In fact we believe that the conceptual simplicity allowed us to apply the framework to a variety of situations leading to algorithms that are potentially amenable to practical implementations. As a first step, we restricted our focus to the above mentioned classical models to capture probabilistic distribution. A natural extension of this work would be to apply our techniques for rigorous verification and testing of neural network models such as Generative Adversarial Networks (GANs) wherein a discriminator is inherently tasked with performing *closeness-testing* for the given data distribution and distribution arising from the generator [LKFO18].

## 2   Previous work

Prior work most related to our work is in the area of distribution testing. The topic of distribution testing is rooted in statistical hypothesis testing and goes back to Pearson's chi-squared test in 1900. In theoretical computers science, distribution testing research is relatively new and focuses on designing hypothesis testers with optimal sample complexity. Goldreich and Ron [GR11] investigated uniformity testing (distinguishing whether an input distribution $P$ is uniform over its support or $\varepsilon$-far from uniform in total variation distance) and designed a tester with sample complexity $O(m/\varepsilon^4)$ (where $m$ is the size of the sample space). Paninski [Pan08] showed that $\Theta(\sqrt{m}/\varepsilon^2)$ samples are necessary for uniformity testing, and gave an optimal tester when $\varepsilon > m^{-1/4}$. Batu et al. [BFR$^+$13] initiated the investigation of identity (goodness-of-fit) testing and closeness (two-sample) testing and gave testers with sample complexity $\tilde{O}(\sqrt{m}/\varepsilon^6)$ and $\tilde{O}(m^{2/3}\mathrm{poly}(1/\varepsilon))$ respectively. Optimal bounds for these testing problems were obtained in Valiant and Valiant [VV17] ($\Theta(\sqrt{m}/\varepsilon^2)$) and Chan et al. [CDVV14] ($\Theta(\max(m^{2/3}\varepsilon^{-4/3}, \sqrt{m}\varepsilon^{-2}))$) respectively. Tolerant versions of these testing problems have very different sample complexity. In particular, Valiant and Valiant [VV11b, VV10] showed that tolerant uniformity, identity, and closeness testing with respect to the total variation distance have a sample complexity of $\Theta(m/\log m)$. Since the seminal papers of Goldreich and Ron and Batu et al., distribution testing grew into a very active research topic and a wide range of properties of distributions have been studied under this paradigm. This research led to sample-optimal testers for many distribution properties. We refer the reader to the surveys [Can15, Rub12] and references therein for more details and results on the topic.

---

[‡]They also present non-tolerant testers for the case when the underlying graph is unknown.

When the sample space is a high-dimensional space (such as $\{0,1\}^n$), the testers designed for general distributions require exponential number of samples ($2^{\Omega(n)}$) if the sample space is $\{0,1\}^n$ for a constant $\varepsilon$). Thus structural assumptions are to be made to design efficient ($\mathrm{poly}(n,1/\varepsilon)$) and practical testers for many of the testing problems. The study of testing high-dimensional distributions with structural restrictions was initiated only very recently. The work that is most closely related to our work appears in [DDK19, CDKS17, DP17, ABDK18] (these works also give good expositions to other prior work on this topic). These papers consider distributions coming from graphical models including Ising models and Bayes nets. In Daskalakis et al. [DDK19], the authors consider distributions that are drawn from an Ising model and show that identity testing and *independence testing* (testing whether an unknown distribution is close to a product distribution) can be done with $\mathrm{poly}(n,1/\varepsilon)$ samples where $n$ is the number nodes in the graph associated with the Ising model. In Canonne et al. [CDKS17] and Daskalakis et al. [DP17], the authors consider identity testing and closeness testing for distributions given by Bayes networks of bounded in-degree. Specifically, they design algorithms with sample complexity $\tilde{O}(2^{3(d+1)/4}n/\varepsilon^2)$ that test closeness of distributions over the same Bayes net with $n$ nodes and in-degree $d$. They also show that $\Theta(\sqrt{n}/\varepsilon^2)$ and $\Theta(\max(\sqrt{n}/\varepsilon^2, n^{3/4}/\varepsilon))$ samples are necessary and sufficient for identity testing and closeness testing respectively of pairs of product distributions (Bayes net with empty graph). Finally, in Acharya et al.[ABDK18], the authors investigate testing problems on *causal Bayesian networks* as defined by Pearl [Pea09] and design efficient ($\mathrm{poly}(n,1/\varepsilon)$) testing algorithms for certain identity and closeness testing problems for them. All these papers consider designing non-tolerant testers and leave open the problem of designing efficient testers that are tolerant for high-dimensional distributions which is the main focus in this paper.

Our main technical result builds on the work of Canonne and Rubinfeld [CR14]. They consider a *dual access model* for testing distributions. In this model, in addition to independent samples, the testing algorithm has also access to an evaluation oracle that gives probability of any item in the sample space. They establish that having access to the evaluation oracle leads to testing algorithms with sample complexity independent of the size of the sample space. Indeed, in order to design testing algorithms, they give an algorithm to additively estimate the total variation distance between two unknown distributions in the dual access model. Our distance estimation algorithm is a direct extension of this algorithm. *Conditional sampling model* has been another related model of interest recently [CFGM16, CRS14, CM19].

**Novelty of our work:** We would like to emphasize that the core conceptual and novel contribution of our work is the establishment of a connection between testing in the dual access model (and in the conditional sampling model) to testing and distance approximation in the standard sampling model. These two models have been investigated separately. Here we use the former results to derive several new efficient tolerant testing algorithms in the standard model for high dimensional distributions, thus extending the state-of-the-art in this area. In this regard, we extend [CR14] to derive Algorithm 1, which in our view is intended to be simple and flexible. We consider the simplicity of Algorithm 1 a core strength of our work.

**Comparison with [CR14]:** Technically, [CR14] assumes a perfect access to the probability mass functions of the two distributions. Instead we work with approximate access to p.m.f.s, the approximation being parameterized by $\beta$ and $\gamma$. In our opinion, this generalization (in Appendix A) does not follow trivially. The usage of approximation has allowed us to obtain results for several high dimensional distributions that do not follow directly from [CR14]. For example, let us consider the Ising model. In this case, given samples from two ferromagnetic Ising models $P$ and $Q$, we approximately learn the model parameters [KM17] and estimate the partition functions [JS93], to evaluate the p.m.f.s approximately. The later result takes parameters of a ferromagnetic Ising model as input and returns a (randomized) PTIME $(1 \pm \gamma)$-multiplicative approximation of its partition function, and therefore we obtain a PTIME algorithm. In contrast, since the computation of the partition function given a fully known ferromagnetic Ising model is known to be #P-complete [JS93] (see Theorem 15 of their paper) and as the algorithm given in [CR14] does not allow for multiplicative errors, directly applying it would lead to an algorithm with $\mathsf{P}^{\#\mathsf{P}}$ complexity. The approximation parameter $\beta$ was used for designing a distance approximation algorithm for all four classes considered in this paper.

## 3 Main Result

We first formalize the connection between learning and distance approximation, and then we give our main algorithm for distance approximation. In the next section, we detail the implications for several well-studied families of structured high-dimensional probability distributions.

Given a family of distributions $\mathcal{D}$, a learning algorithm for $\mathcal{D}$ is an algorithm $\mathcal{L}$ that on input $\varepsilon \in (0, 1)$ and sample access to a distribution $P \in \mathcal{D}$, returns the description of a distribution $\hat{P}$ such that with probability at least $2/3$, $d_{\mathrm{TV}}(P, \hat{P}) \leqslant \varepsilon$.

Our framework for distance approximation needs to (approximately) evaluate the mass function $\hat{P}(x) \coloneqq \mathbf{Pr}_{X \sim \hat{P}}[X = x]$ for any $x \in \Sigma^n$. More precisely, we require EVAL *approximators*:

**Definition 3.1.** *Let $P$ be a distribution over a finite set $U$. A function $E_P : U \to [0, 1]$ is a $(\beta, \gamma)$-EVAL approximator for $P$ if there exists a distribution $\hat{P}$ over $U$ such that*

- $d_{\mathrm{TV}}(P, \hat{P}) \leqslant \beta$

- $\forall x \in U, (1 - \gamma) \cdot \hat{P}(x) \leqslant E_P(x) \leqslant (1 + \gamma) \cdot \hat{P}(x)$

In our applications, we first use a learning algorithm to obtain parameters that describe $\hat{P}$, and then we compute (or approximate) $\hat{P}(x)$ efficiently in terms of these parameters.

**Example 3.2.** Suppose $\mathcal{D}$ is the family of product distributions on $\{0, 1\}^n$. That is, any $P \in \mathcal{D}$ can be described in terms of $n$ parameters $p_1, \ldots, p_n$ where each $p_i$ is the probability of the $i$'th coordinate being 1. It is folklore (see e.g. [ADK15]) that there is a learning algorithm which gets $O(n\varepsilon^{-2})$ samples from $P$ and returns the parameters $\hat{p}_1, \ldots, \hat{p}_n$ of a product distribution $\hat{P}$ satisfying $d_{\mathrm{TV}}(P, \hat{P}) \leqslant \varepsilon$ with probability $2/3$. It is clear that given $\hat{p}_1, \ldots, \hat{p}_n$, we can compute $\hat{P}(x)$ for any $x \in \{0, 1\}^n$ in linear time as: $\hat{P}(x) = \prod_{i=1}^{n} (x_i \cdot \hat{p}_i + (1 - x_i) \cdot (1 - \hat{p}_i))$. Thus, there is an algorithm that takes as input sample access to any product distribution $P$, has sample and time complexity $O(n\varepsilon^{-2})$, and returns a circuit implementing an $(\varepsilon, 0)$-EVAL approximator for $P$. Moreover, any call to the circuit returns in $O(n)$ time.

We establish the following link between EVAL approximators and distance approximation, achieved using Algorithm 1. Its proof can be found in Appendix A.

**Theorem 3.3.** *Suppose we have sample access to distributions $P$ and $Q$ over a finite set. Also, suppose we have access to $(\varepsilon, \varepsilon)$-EVAL approximators for $P$ and $Q$. Then, with probability at least $2/3$, $d_{\mathrm{TV}}(P, Q)$ can be approximated to within $O(\varepsilon)$ additive error using $O(\varepsilon^{-2})$ samples from $P$ and $O(\varepsilon^{-2})$ calls to the two EVAL approximators.*

---

**Algorithm 1:** Distance approximation between $P$ and $Q$

    **Input** : Sample access to distribution $P$; oracle access to $(\varepsilon, \varepsilon)$-EVAL approximators $\mathcal{C}_P$ and $\mathcal{C}_Q$ for $P$ and $Q$ respectively.
    **Output** : Approximate value of $d_{\mathrm{TV}}(P, Q)$
**1 for** $i = 1, \ldots, t = O(\varepsilon^{-2})$ **do**
**2**     Draw a sample $x$ from $P$;
**3**     $a \leftarrow \mathcal{C}_P(x)$;
**4**     $b \leftarrow \mathcal{C}_Q(x)$;
**5**     $c_i \leftarrow \mathbb{1}_{a>b} \left(1 - \frac{b}{a}\right)$;
**6 return** $\frac{1}{t} \sum_{i=1}^{t} c_i$

---

Thus, in the context of Example 3.2, the above theorem immediately implies a distance approximation algorithm for product distributions using $O(n\varepsilon^{-2})$ samples and time. Theorem 3.3 extends the work of Canonne and Rubinfeld [CR14] who considered the setting $\beta = \gamma = 0$. We discussed the relation to prior work in Section 2.

***Testing, learning, and efficiency:*** It is natural to ask whether we can design substantially more efficient distance approximation (or tolerant testing) algorithms than the ones that are possible via

learning as we do in this paper. We discuss this from the perspective of both sample complexity as well as time complexity.

It is clear that the sample complexity of distance approximations is at most that of learning: from the learnt distributions we can compute the distance using a brute-force algorithm (not computationally efficient). On the other hand, current known results give evidence that typically it is not possible to substantially improve the dependence on the dimension ($n$), at least for the following two edge cases. Valiant and Valiant [VV11a] have shown that given samples from an unknown distribution over $[m]$, approximating its distance to the uniform distribution up to a constant additive error with 2/3 probability requires $\Omega(m/\log m)$ samples. In contrast, it is well known that we can learn an unknown distribution within constant error with 2/3 success probability using only $O(m)$ samples. Similarly, in the case of high-dimensional distributions over $\{0,1\}^n$, Canonne et al. [CDKS17] have shown that there exists two *product distributions* whose distance approximation up to a constant error with 2/3 probability requires $\Omega(n/\log n)$ samples, whereas an unknown product distribution can be learnt in constant error with 2/3 probability in $O(n)$ samples. Thus typically sample complexities of learning and distance approximation differ only by a logarithmic factor. However, if one is interested in *non-tolerant* testing, substantial improvements are possible. In particular, for the above problems there are algorithms with $O(\sqrt{m})$ [GR11, Pan08] and $\tilde{O}(\sqrt{n})$ [DP17, CDKS17] sample complexity respectively.

From a time complexity perspective, even if we assume that the learning is perfect, computational efficiency remains a challenge for distance estimation in many high-dimensional settings. Sahai and Vadhan [SV03] have shown that tolerant testing of distributions encoded by Boolean circuits is a problem that is complete for the class SZK (problems admitting statistical zero knowledge interactive proofs). The class SZK contains several hard computational problems including Graph Isomorphism. Kiefer [Kie18] has shown that given two completely specified hidden Markov models, it is #P-hard to additively approximate their distance. Bogdanov et al. [BMV08] have shown that given two completely specified Markov Random Fields with hidden variables, it is impossible to approximate their distance in randomized polynomial time unless NP = RP.

By coupling learning algorithms with the template for distance approximation given by Theorem 3.3, we present a number of scenarios where sample and computational efficient distance approximation algorithms can be designed. We also describe a generic method to efficiently improve the success probability of learning algorithms for the families of distributions admitting a fast distance approximation algorithm, which is presented in Appendix F.

# 4 Applications

## 4.1 Bayesian Networks

A standard way to model structured high-dimensional distributions is through *Bayesian networks*. A Bayesian network describes how a collection of random variables can be generated one-at-a-time in a directed fashion, and they have been used to model beliefs in a wide variety of domains (see [JN07, KF09] for many pointers to the literature). Formally, a probability distribution $P$ over $n$ variables $X_1, \ldots, X_n \in \Sigma$ is said to be a *Bayesian network on a directed acyclic graph $G$ with $n$ nodes* if[§] for every $i \in [n]$, $X_i$ is conditionally independent of $X_{\text{non-descendants}(i)}$ given $X_{\text{parents}(i)}$. Equivalently, $P$ admits the factorization:

$$P(x) \coloneqq \Pr_{X \sim P}[X = x] = \prod_{i=1}^{n} \Pr_{X \sim P}[X_i = x_i \mid \forall j \in \text{parents}(i), X_j = x_j] \qquad \text{for all } x \in \Sigma^n \quad (1)$$

For example, product distributions are Bayesian networks on the empty graph.

Invoking our framework of distance approximation via EVAL approximators on Bayesian networks, we obtain the following:

**Theorem 4.1.** *Suppose $G_1$ and $G_2$ are two DAGs on $n$ vertices with in-degree at most $d$. Let $\mathcal{D}_1$ and $\mathcal{D}_2$ be the family of Bayesian networks on $G_1$ and $G_2$ respectively. Then, there is a distance approximation algorithm for $(\mathcal{D}_1, \mathcal{D}_2)$ that gets $m = \tilde{O}(|\Sigma|^{d+1} n \varepsilon^{-2})$ samples and runs in $O(mn)$ time.*

---

[§]We use the notation $X_S$ to denote $\{X_i : i \in S\}$ for a set $S \subseteq [n]$.

Theorem 4.1 extends the works of Daskalakis et al. [DP17] and Canonne et al. [CDKS17] who designed efficient *non-tolerant* identity and closeness testers for Bayesian networks. Their arguments appear to be inadequate to design tolerant testers. In addition, their results for general Bayesian networks were restricted to the case when $G_1 = G_2$. Theorem 4.1 immediately gives efficient *tolerant* identity and closeness testers for Bayesian networks even when $G_1 \neq G_2$. Canonne et al. [CDKS17] obtain better sample complexity but they make certain *balancedness* assumption on each conditional probability distribution. Without such assumptions, the sample complexity of our algorithm is optimal.

Theorem 4.1 relies on a new learning algorithm for Bayesian networks on a known DAG $G$ that may be of independent interest. It uses $\tilde{O}(n\varepsilon^{-2}|\Sigma|^{d+1})$ samples where $d$ is the maximum in-degree. It returns another Bayesian network $\hat{P}$ on $G$, described in terms of the conditional probability distributions $X_i \mid x_{\text{parents}(i)}$ for all $i \in [n]$ and all settings of $x_{\text{parents}(i)} \in \Sigma^{\deg(i)}$. The sample complexity of the algorithm is nearly optimal. Such a learning algorithm was claimed in the appendix of [CDKS17], but the analysis there appears to be incomplete with no immediate fix [Can20].

## 4.2  Ising Models

Another widely studied model of high-dimensional distributions is the *Ising model*. It was originally introduced in statistical physics as a way to study spin systems ([Isi25]) but has since emerged as a versatile framework to study other systems with pairwise interactions, e.g., social networks ([MS10]), learning in coordination games ([Ell93]), phylogeny trees in evolution ([Ney71, Far73, Cav78]) and image models for computer vision ([GG86]). Formally, a distribution $P$ over variables $X_1, \ldots, X_n \in \{-1, 1\}$ is an *Ising model* if for all $x \in \{-1, 1\}^n$:

$$P(x) = \frac{\exp\left(\sum_{i \neq j \in [n]} A_{ij} x_i x_j + \theta \sum_{i \in [n]} x_i\right)}{\sum_{z \in \{-1,1\}^n} \exp\left(\sum_{i \neq j \in [n]} A_{ij} z_i z_j + \theta \sum_{i \in [n]} z_i\right)} \tag{2}$$

where $\theta \in \mathbb{R}$ is called the *external field* and $A_{ij}$ are called the *interaction terms*. An Ising model is called *ferromagnetic* if all $A_{ij} \geqslant 0$. The *width* of an Ising model as in (2) is $\max_i \sum_j |A_{ij}| + |\theta|$.

Invoking our framework on Ising models, we obtain:

**Theorem 4.2.** *Let $\mathcal{D}$ be the family of ferromagnetic Ising models having width at most $d$. Then, there is a distance approximation algorithm for $\mathcal{D}$ with sample complexity $m = e^{O(d)}\varepsilon^{-4}n^8 \log(\frac{n}{\varepsilon})$ and runtime $O(mn^2 + \varepsilon^{-2}n^{17}\log n)$.*

We use the parameter learning algorithm by Klivans and Meka [KM17] that learns the parameters $\hat{\theta}, \hat{A}_{ij}$ of another Ising model $\hat{P}$ such that $\hat{P}(x)$ is a $(1 \pm \varepsilon)$ approximation of $P(x)$ for every $x$. This results holds for any Ising model, ferromagnetic or not. But in order to get an EVAL approximator, we need to compute $\hat{P}(x)$ from $\hat{\theta}, \hat{A}_{ij}$. In general, the partition function (i.e., the sum in the denominator of Equation (2)) may be #P-hard to compute, but for ferromagnetic Ising models, Jerrum and Sinclair [JS93] gave a PTAS for this problem. Thus, we obtain an $(\varepsilon, \varepsilon)$-EVAL approximator for ferromagnetic Ising models that runs in polynomial time, and then Theorem 4.2 follows from Theorem 3.3.

Daskalakis et al. [DDK19] studied independence testing and identity testing for Ising models and design *non-tolerant* testers. Their sample and time complexity have polynomial dependence on the width instead of exponential (as in our case), but their algorithms seem to be inherently non-tolerant. In contrast, our distance approximation algorithm leads to a tolerant closeness-testing algorithm for ferromagnetic Ising models. Also, Theorem 4.2 offers a template for distance approximation algorithms whenever the partition function can be approximated efficiently. In particular, Sinclair et al [SST14] showed a PTAS for computing the partition function of anti-ferromagnetic Ising models in certain parameter regimes.

We also show that we can efficiently approximate the distance to uniformity for any Ising model.

**Theorem 4.3.** *There is an algorithm which, given independent samples from an unknown Ising model $P$ over $\{-1, 1\}^n$ with width at most $d$, takes $m = O(e^{O(d)}\varepsilon^{-4}n^8 \log(n/\varepsilon))$ samples, $O(mn^2)$ time and returns a value $e$ such that $|e - d_{\text{TV}}(P, U)| \leqslant \varepsilon$ with probability at least 7/12, where $U$ is the uniform distribution over $\{-1, 1\}^n$.*

The proof of Theorem 4.3 proceeds by learning the parameters $\hat{\theta}, \hat{A}$ of an Ising model $\hat{P}$ that is a multiplicative approximation fo $P$. As we mentioned earlier, computing the partition function is in general hard. However, we can efficiently estimate the ratio $P(x)/P(y)$ for any two $x, y \in \{-1, 1\}^n$. At this point, we invoke the uniformity tester by Narayanan [Nar21] that uses samples from the input distribution as well as pairwise conditional samples (in the PCOND oracle model).

## 4.3 Multivariate Gaussians

Theorem 3.3 applies also when the sample space is not finite, e.g., the reals. Then, in the definition of the $(\beta, \gamma)$-EVAL approximator $E_P$ for a distribution $P$, we require a distribution $\hat{P}$ such that $d_{\text{TV}}(P, \hat{P}) \leqslant \beta$ and $E_P$ is a $(1 \pm \gamma)$-approximation of the *probability density function* of $\hat{P}$ at any $x$.

The most prominent instance in which we can apply our framework in this setting is for the class of multivariate gaussians, again another widely used model for high-dimensional distributions used throughout the natural and social sciences (see, e.g., [MDLW18]). There are two main reasons for their ubiquity. Firstly, because of the central limit theorem, any physical quantity that is a population average is approximately distributed as a gaussian. Secondly, the gaussian distribution has maximum entropy among all real-valued distributions with a particular mean and covariance; therefore, a gaussian model places the least restrictions beyond the first and second moments of the distribution.

For $\mu \in \mathbb{R}^n$ and positive definite $\Sigma \in \mathbb{R}^{n \times n}$, the distribution $N(\mu, \Sigma)$ has the density function:

$$N(\mu, \Sigma; x) = \frac{1}{(2\pi)^{n/2}\sqrt{\det(\Sigma)}} \exp\left(-\frac{1}{2}(x-\mu)^\top \Sigma^{-1}(x-\mu)\right) \tag{3}$$

Invoking our framework on multivariate gaussians, we obtain:

**Theorem 4.4.** *Let $\mathcal{D}$ be the family of multivariate gaussian distributions, $\{N(\mu, \Sigma) : \mu \in \mathbb{R}^n, \Sigma \in \mathbb{R}^{n \times n}, \Sigma \succ 0\}$. Then, there is a distance approximation algorithm for $\mathcal{D}$ with sample complexity $O(n^2 \varepsilon^{-2})$ and runtime $O(n^\omega \varepsilon^{-2})$ (where $\omega > 2$ is the matrix multiplication constant).*

It is folklore (see [ABDH+17] for a proof) that for any $P = N(\mu, \Sigma)$, the empirical mean $\hat{\mu}$ and empirical covariance $\hat{\Sigma}$ obtained from $O(n^2 \varepsilon^{-2})$ samples from $P$ determines a gaussian $\hat{P} = N(\hat{\mu}, \hat{\Sigma})$ satisfying $d_{\text{TV}}(P, \hat{P}) \leqslant \varepsilon$ with probability at least 3/4. To get an EVAL approximator, we need evaluations of $N(\hat{\mu}, \hat{\Sigma}; x)$ for any $x$ as in (3). Since $\det(\hat{\Sigma})$ is computable in time $O(n^\omega)$, Theorem 4.4 follows from Theorem 3.3.

This result is interesting because there is no closed-form expression known for the total variation distance between two gaussians of specified mean and covariance. Devroye et al. [DMR18] give expressions for lower- and upper-bounding the total variation distance that are a constant multiplicative factor away from each other. On the other hand, our approach yields a polynomial time randomized algorithm that, given $\mu_1, \Sigma_1, \mu_2, \Sigma_2$, approximates the total variation distance $d_{\text{TV}}(N(\mu_1, \Sigma_1), N(\mu_2, \Sigma_2))$ upto $\pm\varepsilon$ additive error.

**Corollary 4.5.** *For any two vectors $\mu_1, \mu_2 \in \mathbb{R}^n$ and two positive-definite matrices $\Sigma_1, \Sigma_2 \in \mathbb{R}^{n \times n}$, $d_{\text{TV}}(N(\mu_1, \Sigma_1), N(\mu_1, \Sigma_1))$ can be estimated up to an additive $\varepsilon$ error in $O(n^3 \varepsilon^{-2})$ time.*

*Proof.* We again invoke Algorithm 2. Since the parameters are already provided, we can readily obtain $(0, 0)$-EVAL approximators for $N(\mu_1, \Sigma_1)$ and $N(\mu_2, \Sigma_2)$. For Algorithm 2, we also need sample access to one of the two distributions. It is well known that if $v \sim N(0, I)$ and $\Sigma = LL^\top$, then $Lv + \mu \sim N(\mu, \Sigma)$; the matrix $L$ can be obtained in $O(n^3)$ time using a Cholesky decomposition. Hence, each sample from $N(\mu_1, \Sigma_1)$ costs $O(n^3)$ time, so that the entire algorithm runs in $O(n^3 \varepsilon^{-2})$ time. $\square$

## 4.4 Interventional Distributions in Causal Models

A *causal model* for a system of random variables describes not only how the variables are correlated but also how they would change if they were to be externally set to prescribed values. To formalize this, we can use the language of *causal Bayesian networks* due to Pearl [Pea09]. A causal Bayesian network is a Bayesian network with an extra *modularity* assumption: for each node $i$ in the network, the dependence of $X_i$ on $X_{\text{parents}(i)}$ is an autonomous mechanism that does not change even if other parts of the network are changed.

Suppose $\mathcal{P}$ is a causal Bayesian network over variables $X_1, \ldots, X_n$ on a directed acyclic graph $G$ with nodes labeled $\{1, \ldots, n\}$. The nodes in $G$ are partitioned into two sets: *observable $V$* and *hidden $U$*. A sample from the observational distribution $P$ yields the values of variables $X_V = \{X_i : \in V\}$. The modularity assumption allows us to define the result of *interventions* on causal Bayesian networks. An intervention is specified by a subset $S \subseteq V$ and an assignment $s \in \Sigma^{|S|}$. In the resulting interventional distribution, the variables in $S$ are fixed to $s$, while the variables $X_i$ for $i \notin S$ are sampled in topological order as it would have been in the original Bayesian network, according to the conditional probability distribution $X_i \mid X_{\mathrm{parents}(i)}$, where $X_{\mathrm{parents}(i)}$ consist of either variables previously sampled in the topological order or variables in $S$ set by the intervention. Finally, the variables in $U$ are marginalized out. The resulting distribution on $X_V$ is denoted $P_s$.

The question of inferring the interventional distribution from samples is a fundamental one. We focus on *atomic interventions*, i.e., where the intervention is on a single node $A \in V$. In this case, Tian and Pearl [TP02a, Tia02] exactly characterized the graphs $G$ for which any causal Bayesian network $\mathcal{P}$ on $G$ and for any assignment $a \in \Sigma$ to $X_A$, the interventional distribution $P_a$ is *identifiable* from the observational distribution $P$ on $X_V$. For identification to be computationally effective, it is also natural to require certain *strong positivity* condition on $P$. We show that we can efficiently estimate the distances between interventional distributions of causal Bayesian networks whenever the identifiability and strong positivity conditions are met. See Appendix E for necessary definitions.

**Theorem 4.6** (Informal). *Suppose $\mathcal{P}, \mathcal{Q}$ are two unknown causal Bayesian networks on two known graphs $G_1$ and $G_2$ on a common observable set $V$ containing a special node $A$ and having bounded in-degree and c-component size. Suppose $G_1$ and $G_2$ both satisfy the identifiability condition, and the observational distributions $P$ and $Q$ satisfy the strong positivity condition. Then there is an algorithm which for any $a \in \Sigma$ and parameter $\varepsilon \in (0, 1)$ returns a value $e$ such that $|e - d_{\mathrm{TV}}(P_a, Q_a)| \leqslant \varepsilon$ with probability at least 2/3 using $\mathrm{poly}(|\Sigma|, n, \varepsilon^{-1})$ samples from the observational distributions $P$ and $Q$ and running in time $\mathrm{poly}(|\Sigma|, n, \varepsilon^{-1})$.*

We again use the framework of EVAL approximators to prove the theorem, but there is a complication: we do not get samples from the distributions $P_a$ and $Q_a$, but only from $P$ and $Q$. We build on a recent work ([BGK$^+$20]) that shows how to efficiently learn and sample from interventional distributions of atomic interventions using observational samples.

Theorem 4.6 solves a natural problem. Suppose a biologist wants to compare how a particular point mutation affects the activity of other genes for Africans and for Europeans. Because of ethical reasons, she cannot conduct randomized controlled trials by actively inducing the mutation, but she can draw random samples from the two populations. It is reasonable to assume that the graph structure of the regulatory network is the same for all individuals and that the causal graph over the genes of interest is known (or can be learned through other methods). Also, suppose that the gene expression levels can be discretized. She can then, in principle, use the algorithm proposed in Theorem 4.6 to test whether the effect of the mutation is approximately the same for Africans and Europeans.

## 4.5 Tightness of Our Bounds

In this paper our focus was mainly establishing upper bounds. We note that the $\Omega(\frac{n}{\log n})$ lower bound from [CDKS17] mentioned earlier for tolerant testing of product distributions, implies the same lower bound for tolerant testing of Bayes nets and atomic interventional distributions. For the Ising model, currently we do not have a lower bound in general.

## Broader Impact

This work presents basic algorithms for approximating distances between two high dimensional distributions. While the results are theoretical in nature and do not present any immediate societal consequences, the algorithms have potential to impact practice in the long term.

**Acknowledgement**   We thank the anonymous reviewers of NeurIPS20 for their valuable suggestions for improving our paper. This work was supported in part by National Research Foundation Singapore under its NRF Fellowship Programme [NRF-NRFFAI1-2019-0002, NRF-NRFFAI1-2019-0004] and AI Singapore Programme [AISG-RP-2018-005], NUS ODPRT Grant [R-252-000-685-13], and NUS ODPRT Grant [R-252-000-A33133]. Any opinions, findings and conclusions or recommendations expressed in this material are those of the author(s) and do not reflect the views of National Research Foundation, Singapore. The work of Bhattacharyya was additionally supported by an Amazon Research Award. The work of Vinodchandran was supported in part by the US National Science Foundation under grants NSF CCF-184908 and NSF HDR:TRIPODS-1934884. All opinions are of the authors and do not reflect the view of sponsors.

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
