[Supplementary Material]

# A Distance Approximation Algorithm

In this section, we prove Theorem 3.3 which underlies all the other results in this work. In fact, we show the following theorem that is more detailed.

**Theorem A.1.** *Suppose we have sample access to distributions $P$ and $Q$ over a finite set. Also, suppose we can make calls to two circuits $\mathcal{C}_P$ and $\mathcal{C}_Q$ which implement $(\beta, \gamma)$-EVAL approximators for $P$ and $Q$ respectively. Let $T$ be the maximum running time for any call to $\mathcal{C}_P$ or $\mathcal{C}_Q$.*

*Then for any $\varepsilon, \delta > 0$, $d_{\mathrm{TV}}(P, Q)$ can be approximated up to an additive error $\frac{2\gamma}{1-\gamma} + 3\beta + \varepsilon$ with probability at least $1 - \delta$, using $O(\varepsilon^{-2} \log \delta^{-1})$ samples from $P$ and $O(\varepsilon^{-2} \log \delta^{-1} \cdot T)$ runtime.*

Note that the EVAL approximators in Theorem A.1 must return rational numbers with bounded denominators as they are implemented by circuits with bounded running time. The exact model of computation for the circuits does not matter so much, so we omit its discussion.

We now turn to the proof of Theorem A.1. As mentioned in the Introduction, if $\mathcal{C}_P$ and $\mathcal{C}_Q$ were $(0,0)$-EVAL approximators, the result already appears in [CR14]. The proof below analyzes how having nonzero $\beta$ and $\gamma$ affects the error bound.

---

**Algorithm 2:** Distance approximation

> **Input** : Sample access to distribution $P$; oracle access to circuits $\mathcal{C}_P$ and $\mathcal{C}_Q$.
> **Output** : Approximate value of $d_{\mathrm{TV}}(P, Q)$
> **1 for** $i = 1, \ldots, t = O(\varepsilon^{-2} \log \delta^{-1})$ **do**
> **2** $\quad$ Draw a sample $x$ from $P$;
> **3** $\quad$ $a \leftarrow \mathcal{C}_P(x)$;
> **4** $\quad$ $b \leftarrow \mathcal{C}_Q(x)$;
> **5** $\quad$ $c_i \leftarrow \mathbb{1}_{a>b}\left(1 - \frac{b}{a}\right)$;
> **6 return** $\frac{1}{t} \sum_{i=1}^{t} c_i$

---

*Proof.* We invoke Algorithm 2. Notice that the algorithm only requires sample access to one of the two distributions but to both of the EVAL approximators. Let $\hat{P}$ be the distribution $\beta$-close to $P$ which is approximated by the output of $\mathcal{C}_P$; similarly define $\hat{Q}$.

We have $|d_{\mathrm{TV}}(P, Q) - d_{\mathrm{TV}}(\hat{P}, \hat{Q})| \leqslant d_{\mathrm{TV}}(P, \hat{P}) + d_{\mathrm{TV}}(Q, \hat{Q}) \leqslant 2\beta$ from the triangle inequality. Hence, it is sufficient to approximate $d_{\mathrm{TV}}(\hat{P}, \hat{Q})$ additively up to $\frac{2\gamma}{1-\gamma} + \beta + \varepsilon$.

$$
\begin{aligned}
d_{\mathrm{TV}}(\hat{P}, \hat{Q}) &= \frac{1}{2} \sum_x |\hat{P}(x) - \hat{Q}(x)| \\
&= \sum_{x: \hat{P}(x) > \hat{Q}(x)} (\hat{P}(x) - \hat{Q}(x)) \\
&= \sum_{x: \hat{P}(x) > \hat{Q}(x)} \left(1 - \frac{\hat{Q}(x)}{\hat{P}(x)}\right) \hat{P}(x) \qquad \text{(Since } \hat{P}(x) > 0\text{)} \\
&= \mathop{\mathbf{E}}_{x \sim \hat{P}} \left[ \mathbb{1}_{\hat{P}(x) > \hat{Q}(x)} \left(1 - \frac{\hat{Q}(x)}{\hat{P}(x)}\right) \right]
\end{aligned}
$$

From the above, if we have complete access (both evaluation and sample) to $\hat{P}$ and $\hat{Q}$, then we can estimate the distance with $O(\frac{1}{\varepsilon^2} \log \frac{1}{\delta})$ samples and evaluations. However as we have only approximate evaluations of $\hat{P}$ and $\hat{Q}$ and samples from the original distribution $P$, we need some additional arguments. Let $E_P$ and $E_Q$ be the functions implemented by the circuits $\mathcal{C}_P$ and $\mathcal{C}_Q$ respectively.

$$d_{\mathrm{TV}}(\hat{P}, \hat{Q}) = \sum_x 1_{\hat{P}(x) > \hat{Q}(x)} \left(1 - \frac{\hat{Q}(x)}{\hat{P}(x)}\right) \hat{P}(x)$$

$$= \underbrace{\sum_x 1_{E_P(x) > E_Q(x)} \left(1 - \frac{E_Q(x)}{E_P(x)}\right) \hat{P}(x)}_{A} +$$

$$\underbrace{\sum_x \left[1_{\hat{P}(x) > \hat{Q}(x)} \left(1 - \frac{\hat{Q}(x)}{\hat{P}(x)}\right) - 1_{E_P(x) > E_Q(x)} \left(1 - \frac{E_Q(x)}{E_P(x)}\right)\right] \hat{P}(x)}_{B}$$

We start with an upper bound for the absolute value of the error term $B$. We consider the partition of sample space into $S_1$, $S_2$ and $S_3$, where $S_1 = \{x : 1_{\hat{P}(x) > \hat{Q}(x)} = 1_{E_P(x) > E_Q(x)}\}$, $S_2 = \{x : 1_{\hat{P}(x) > \hat{Q}(x)} > 1_{E_P(x) > E_Q(x)}\}$ and $S_3 = \{x : 1_{\hat{P}(x) > \hat{Q}(x)} < 1_{E_P(x) > E_Q(x)}\}$.

$$|B| = \left|\sum_x \left[1_{\hat{P}(x) > \hat{Q}(x)} \left(1 - \frac{\hat{Q}(x)}{\hat{P}(x)}\right) - 1_{E_P(x) > E_Q(x)} \left(1 - \frac{E_Q(x)}{E_P(x)}\right)\right] \hat{P}(x)\right|$$

$$\leqslant \sum_x \left|\left[1_{\hat{P}(x) > \hat{Q}(x)} \left(1 - \frac{\hat{Q}(x)}{\hat{P}(x)}\right) - 1_{E_P(x) > E_Q(x)} \left(1 - \frac{E_Q(x)}{E_P(x)}\right)\right] \hat{P}(x)\right|$$

$$= \sum_{x \in S_1} 1_{\hat{P}(x) > \hat{Q}(x)} \left|\frac{\hat{Q}(x)}{\hat{P}(x)} - \frac{E_Q(x)}{E_P(x)}\right| \hat{P}(x) + \sum_{x \in S_2} 1_{\hat{P}(x) > \hat{Q}(x)} \left(1 - \frac{\hat{Q}(x)}{\hat{P}(x)}\right) \hat{P}(x) +$$

$$\sum_{x \in S_3} 1_{E_P(x) > E_Q(x)} \left(1 - \frac{E_Q(x)}{E_P(x)}\right) \hat{P}(x)$$

For $x$ in $S_1$ with $\hat{P}(x) > \hat{Q}(x)$, $\frac{(1-\gamma)}{(1+\gamma)} \frac{\hat{Q}(x)}{\hat{P}(x)} \leqslant \frac{E_Q(x)}{E_P(x)} \leqslant \frac{(1+\gamma)}{(1-\gamma)} \frac{\hat{Q}(x)}{\hat{P}(x)}$ so that $\left|\frac{\hat{Q}(x)}{\hat{P}(x)} - \frac{E_Q(x)}{E_P(x)}\right| \leqslant \frac{2\gamma}{1-\gamma} \frac{\hat{Q}(x)}{\hat{P}(x)} < \frac{2\gamma}{1-\gamma}$. For $x$ in $S_2$, $\hat{P}(x) > \hat{Q}(x)$ implies $E_P(x) \leqslant E_Q(x)$ and hence, $(1-\gamma)\hat{P}(x) \leqslant E_P(x) \leqslant E_Q(x) \leqslant (1+\gamma)\hat{Q}(x)$ so that $\hat{Q}(x)/\hat{P}(x) \geqslant \frac{1-\gamma}{1+\gamma}$. For $x$ in $S_3$, $E_P(x) > E_Q(x)$ implies $\hat{P}(x) \leqslant \hat{Q}(x)$, and hence, $\frac{E_Q(x)}{E_P(x)} \geqslant \frac{(1-\gamma)\hat{Q}(x)}{(1+\gamma)\hat{P}(x)} \geqslant \frac{1-\gamma}{1+\gamma}$. Therefore:

$$|B| \leqslant \sum_{x \in S_1} \frac{2\gamma}{1-\gamma} \hat{P}(x) + \sum_{x \in S_2} \frac{2\gamma}{1+\gamma} \hat{P}(x) + \sum_{x \in S_3} \frac{2\gamma}{1+\gamma} \hat{P}(x)$$

$$\leqslant \frac{2\gamma}{1-\gamma}$$

Now consider the term $A$:

$$A = \sum_x 1_{E_P(x) > E_Q(x)} \left(1 - \frac{E_Q(x)}{E_P(x)}\right) \hat{P}(x)$$

$$= \underbrace{\sum_x 1_{E_P(x) > E_Q(x)} \left(1 - \frac{E_Q(x)}{E_P(x)}\right) P(x)}_{C} + \sum_x 1_{E_P(x) > E_Q(x)} \left(1 - \frac{E_Q(x)}{E_P(x)}\right) (\hat{P}(x) - P(x)).$$

Note that: $\left|\sum_x 1_{E_P(x) > E_Q(x)} \left(1 - \frac{E_Q(x)}{E_P(x)}\right) (\hat{P}(x) - P(x))\right| \leqslant \sum_x |\hat{P}(x) - P(x)| \leqslant \beta$. So, $|d_{\mathrm{TV}}(\hat{P}, \hat{Q}) - C| \leqslant \frac{2\gamma}{1-\gamma} + \beta$. We can rewrite $C$ as $\mathbf{E}_{x \sim P} \left[1_{E_P(x) > E_Q(x)} \left(1 - \frac{E_Q(x)}{E_P(x)}\right)\right]$. Since $1_{E_P(x) > E_Q(x)} \left(1 - \frac{E_Q(x)}{E_P(x)}\right)$ lies in $[0, 1]$, by the Hoeffding bound, we can estimate the expectation up to $\varepsilon$ additive error with probability at least $(1 - \delta)$ by averaging $O(\frac{1}{\varepsilon^2} \log \frac{1}{\delta})$ samples from $P$. $\quad\square$

Theorem A.1 can be extended to the case that $P$ and $Q$ are distributions over $\mathbb{R}^n$ with infinite support. We change Definition 3.1 so that $E_P(x)$ is a $(1 \pm \gamma)$-approximation of $\hat{f}(x)$ where $\hat{f}(x)$ is the probability density function for $\hat{P}$. Then, Theorem A.1 and Algorithm 2 continue to hold as stated. In the proof, we merely have to replace the summations with the appropriate integrals.

## B  Bayesian networks

First we apply our distance estimation algorithm for tolerant testing of high dimensional distributions coming from bounded in-degree Bayesian networks. Bayesian networks defined below are popular probabilistic graphical models for describing high-dimensional distributions succinctly.

**Definition B.1.** *A Bayesian network $P$ on a directed acyclic graph $G$ over the vertex set $[n]$ is a joint distribution of the $n$ random variables $(X_1, X_2, \ldots, X_n)$ over the sample space $\Sigma^n$ such that for every $i \in [n]$ $X_i$ is conditionally independent of $X_{\text{non-descendants}(i)}$ given $X_{\text{parents}(i)}$, where for $S \subseteq [n]$, $X_S$ is the joint distribution of $(X_i : i \in S)$, and parents and non-descendants are defined from $G$.*

*$P$ factorizes as follows:*

$$P(x) \coloneqq \Pr_{X \sim P}[X = x] = \prod_{i=1}^{n} \Pr_{X \sim P}[X_i = x_i \mid \forall j \in \text{parents}(i), X_j = x_j] \qquad \text{for all } x \in \Sigma^n \quad (4)$$

*Hence a Bayesian network can be completely described by a set of conditional distributions for every variable $X_i$, for every fixing of its parents $X_{\text{parents}(i)}$.*

To construct an EVAL approximator for a Bayesian network, we first learn it using an efficient algorithm. We show the following proper learning algorithm for Bayesian networks that uses near-optimal sample complexity [CDKS17].

**Theorem B.2.** *There is an algorithm that given a parameter $\varepsilon > 0$ and sample access to an unknown Bayesian network distribution $P$ on a known directed acyclic graph $G$ of in-degree at most $d$, returns a Bayesian network $\hat{P}$ on $G$ such that $d_{\text{TV}}(P, \hat{P}) \leqslant \varepsilon$ with probability $\geqslant 9/10$. Letting $\Sigma$ denote the range of each variable $X_i$, the algorithm takes $m = O(|\Sigma|^{d+1} n \log(|\Sigma|^{d+1} n) \varepsilon^{-2})$ samples and runs in $O(mn)$ time.*

This directly gives us a distance estimation algorithm for Bayesian networks.

**Theorem 4.1.** *Suppose $G_1$ and $G_2$ are two DAGs on $n$ vertices with in-degree at most $d$. Let $\mathcal{D}_1$ and $\mathcal{D}_2$ be the family of Bayesian networks on $G_1$ and $G_2$ respectively. Then, there is a distance approximation algorithm for $(\mathcal{D}_1, \mathcal{D}_2)$ that gets $m = \tilde{O}(|\Sigma|^{d+1} n \varepsilon^{-2})$ samples and runs in $O(mn)$ time.*

*Proof.* Given samples from $P_1$ and $P_2$ we first learn them as $\hat{P}_1$ and $\hat{P}_2$ using Theorem B.2 in $d_{\text{TV}}$ distance $\varepsilon/4$. This step costs $m = O(|\Sigma|^{d+1} n \log(|\Sigma|^{d+1} n) \varepsilon^{-2})$ samples and $O(|\Sigma|^{d+1} mn)$ time and succeeds with probability $4/5$. $\hat{P}_1$ and $\hat{P}_2$ gives efficient $(\varepsilon/4, 0)$-EVAL approximators from Equation (4). It follows from Theorem A.1 that we can estimate $d_{\text{TV}}(P_1, P_2)$ up to an $\varepsilon$ additive error using $O(\varepsilon^{-2})$ additional samples from $P_1$ except for $1/5$ probability. $\qquad \square$

Regarding lower bounds, Canonne et al. [CDKS17] have shown a lower bound of $\Omega(n/\log n)$ samples for deciding for two product distributions $P$ and $Q$ over $\{0,1\}^n$, whether $d_{\text{TV}}(P, Q) \leqslant \varepsilon_0$ versus $d_{\text{TV}}(P, Q) \geqslant 2\varepsilon_0$ with probability $2/3$ for a constant $\varepsilon_0$. On the other hand, Daskalakis et al. [DDK19] have shown that there exists an unknown Bayes net $P$ over $\{0,1\}^n$ whose underlying graph is unknown but known to be a tree such that deciding $d_{\text{TV}}(P, U) = 0$ versus $d_{\text{TV}}(P, U) \geqslant \varepsilon$ with $2/3$ probability requires $\Omega(n\varepsilon^{-2})$ samples, where $U$ is the uniform distribution over $\{0,1\}^n$.

### B.1  Learning Bayesian networks

In this section, we prove a strengthened version of Theorem B.2 that holds for any desired error probability $\delta$.

**Theorem B.3.** *There is an algorithm that given parameters $\varepsilon, \delta > 0$ and sample access to an unknown Bayesian network distribution $P$ on a known directed acyclic graph $G$ of in-degree at most $d$, returns a Bayesian network $Q$ on $G$ such that $d_{\mathrm{TV}}(P, Q) \leqslant \varepsilon$ with probability $\geqslant (1 - \delta)$. Letting $\Sigma$ denote the alphabet for each variable $X_i$, the algorithm takes $m = O(|\Sigma|^{d+1} n \log(|\Sigma|^{d+1} n) \varepsilon^{-2} \log \frac{1}{\delta})$ samples and runs in $O(mn \log^2 \frac{1}{\delta})$ time.*

We actually prove a stronger bound on the distance between $P$ and $Q$ in terms of the KL divergence. The KL divergence between two distributions $P$ and $Q$ is defined as $\mathrm{KL}(P, Q) = \sum_i P(i) \ln \frac{P(i)}{Q(i)}$. From Pinsker's inequality, we have $d_{\mathrm{TV}}{}^2(P, Q) \leqslant 2\mathrm{KL}(P, Q)$. Thus a $d_{\mathrm{TV}}$ learning result follows from a KL learning result. We present Algorithm 3 for the binary alphabet case ($\Sigma = \{0, 1\}$) and reduce the general case to the binary case afterwards.

The *add-1 empirical estimator* takes $z$ samples from a distribution over $k$ items and assigns to item $i$ the probability $(z_i + 1)/(z + k)$ where $z_i$ is the number of occurrences of item $i$ in the samples. We will use the following general result for learning a distribution in KL distance.

**Theorem B.4** ([KOPS15]). *Let $D$ be an unknown distribution over $k$ items. Let $\hat{D}$ be the add-1 empirical distribution of $z$ samples from $D$. Then for $k \geqslant 2$, $z \geqslant 1$, $\mathbf{E}[\mathrm{KL}(D, \hat{D})] \leqslant (k-1)/(z+1)$.*

We will use a KL local additivity result for Bayesian networks, a proof of which is given in [CDKS17]. For a Bayesian network $P$, a vertex $i$, and a setting a value $a$ of its parents, let $\Pi[i, a]$ denote the event that parents of $i$ take value $a$, and let $P(i \mid a)$ denote the distribution at vertex $i$ when its parents takes value $a$.

**Theorem B.5.** *Let $P$ and $Q$ be two Bayesian networks over the same graph $G$. Then*

$$\mathrm{KL}(P, Q) = \sum_i \sum_a P[\Pi[i, a]] \cdot \mathrm{KL}(P(i \mid a), Q(i \mid a))$$

---

**Algorithm 3:** Fixed-structure Bayesian network learning

> **Input** : Samples from an unknown Bayesian network $P$ over $\{0, 1\}^n$ on a known graph $G$ of in-degree $\leqslant d$, parameters $m, t$
> **Output** : A Bayesian network $Q$ over $G$

1 Get $m$ samples from $P$;
2 **for** *every vertex $i$* **do**
3      **for** *every fixing $a$ of $i$'s parents* **do**
4          $N_{i,a} \leftarrow$ the number of samples where $i$'s parents are set to $a$;
5          **if** $N_{i,a} \geqslant t$ **then**
6              $Q(i \mid a) \leftarrow$ the add-1 empirical distribution at node $i$ in the subset of samples where $i$'s parents are set to $a$;
7          **else**
8              $Q(i \mid a) \leftarrow$ uniformly random bit;

---

**Lemma B.6.** *For $m = 24n2^d \log(n2^d)/\varepsilon$ and $t = 12 \log(n2^d)$, Algorithm 3 satisfies $\mathrm{KL}(P, Q) \leqslant 5\varepsilon$ with probability at least 3/4 over the randomness of sampling.*

*Proof.* Call a tuple $(i, a)$ *heavy* if $P[\Pi[i, a]] \geqslant \frac{\varepsilon}{2^d n}$ and *light* otherwise. Let $N_{i,a}$ denote the number of samples where $i$'s parents are $a$.

For every heavy $(i, a)$, let $F_{i,a}$ be the event "$N_{i,a} \geqslant n2^d P[\Pi[i, a]] t/\varepsilon$" and $G_{i,a} = \bigwedge_{\substack{(j,b) \text{ heavy} \\ (j,b) \neq (i,a)}} F_{(j,b)}$. Let $F = G_{i,a} \wedge F_{i,a}$. It is easy to see from Chernoff and union bounds that $F$ is true with 19/20 probability. Hence for the rest of the argument, we condition on this event. In this case, all heavy items satisfy $N_{i,a} \geqslant t$.

Then for any random variable $X$, $\mathbf{E}[X \mid F_{i,a}] = \mathbf{E}[X \mid F]\Pr[G_{i,a} \mid F_{i,a}] + \mathbf{E}[X \mid F_{i,a} \wedge \overline{G_{i,a}}]\Pr[\overline{G_{i,a}} \mid F_{i,a}]$. Hence, $\mathbf{E}[X \mid F] \leqslant \frac{20}{19} \mathbf{E}[X \mid F_{i,a}]$. Similarly, $\mathbf{E}[X \mid F] \leqslant \frac{20}{19} \mathbf{E}[X]$.

Now, we see that:

– For any heavy $(i, a)$, by Theorem B.4,

$$\mathbf{E}[\mathrm{KL}(P(i \mid a), Q(i \mid a)) \mid F_{i,a}] \leqslant \frac{\varepsilon}{12n2^d \cdot P[\Pi[i, a]]}.$$

– Similarly, for any light $(i, a)$ that satisfies $N_{i,a} \geqslant t$, it follows from Theorem B.4 that $\mathbf{E}[\mathrm{KL}(P(i \mid a), Q(i \mid a)) \mid N_{i,a} \geqslant t] \leqslant \frac{1}{12}$.

– Items which do not satisfy $N_{i,a} \geqslant t$ must be light for which $[\mathrm{KL}(P(i \mid a), Q(i \mid a)) \mid N_{i,a} < t] \leqslant p \ln 2p + (1 - p) \ln 2(1 - p) \leqslant \ln 2$, where $p = P[i = 1 | a]$, since in that case $Q(i \mid a)$ is the uniformly random bit.

Using Theorem B.5, we get

$$\mathbf{E}[\mathrm{KL}(P, Q) \mid F] \leqslant \frac{20}{19} \left[ \sum_{(i,a) \text{ heavy}} P[\Pi[i, a]] \cdot \frac{\varepsilon}{12n2^d \cdot P[\Pi[i, a]]} + \sum_{(i,a) \text{ light}} \frac{\varepsilon}{n2^d} \ln 2 \right] \leqslant \varepsilon.$$

The lemma follows from Markov's inequality. $\qquad \square$

Now we reduce the case when $\Sigma$ is not binary to the binary case. We can encode each $\sigma \in \Sigma$ of the Bayesian network as a $\log |\Sigma|$ size boolean string which gives us a Bayesian network of degree $(d + 1) \log |\Sigma|$ over $n \log |\Sigma|$ variables. Then we apply Lemma B.6 to get a learning algorithm with $O(\varepsilon)$ error in $d_{\mathrm{TV}}$ and 3/4 success probability. Subsequently we repeat $O(\log \frac{1}{\delta})$ times and find out a successful repetition using Theorem F.1.

## C    Ising Models

In this section, we give a distance approximation algorithm for the class of bounded-width ferromagnetic Ising models. Recall from Section 4.2 that a probability distribution $P$ from this class is over the sample space $\{-1, 1\}^n$ and that $P(x)$, the probability of an item $x \in \{-1, 1\}^n$, is proportional to the numerator:

$$N(x) = \exp \left( \sum_{i,j} A_{i,j} x_i x_j + \theta \sum_i x_i \right),$$

where $A_{i,j}$s and $\theta$ are parameters of the model. The constant of proportionality, also called the *partition function* of the Ising model is $Z = \sum_x N(x)$, which gives $P(x) = N(x)/Z$. The *width* of the Ising model is defined as $\max_i \sum_j |A_{i,j}| + \theta$. In a *ferromagnetic* Ising model, each $A_{i,j} \geqslant 0$.

Given two such Ising models, we give an algorithm for additively estimating their total variation distance. We first learn these two Ising models up to total variation distance $\varepsilon/8$ using the following learning algorithm given by Klivans and Meka [KM17]. In fact, it gives a stronger $(1 \pm \varepsilon)$ multiplicative approximation guarantee for every probability value.

**Theorem C.1** (Theorem 7.3 in [KM17])**.** *There is an algorithm which, given independent samples from an unknown Ising model $P$ with width at most $d$, returns parameters $\hat{A}_{i,j}$ and $\hat{\theta}$ such that the Ising model $\hat{P}$ constructed with the latter parameters satisfies $(1 - \varepsilon)P(x) \leqslant \hat{P}(x) \leqslant (1 + \varepsilon)P(x)$ for all $x \in \{-1, 1\}^n$. This algorithm takes $m = e^{O(d)} \varepsilon^{-4} n^8 \log(n/\delta\varepsilon)$ samples, $O(mn^2)$ time and succeeds with probability $1 - \delta$.*

However learning the parameters of an Ising model is not enough to efficiently evaluate the probability at arbitrary points. Naively computing the constant of proportionality $Z$ would take $2^n$ time. For certain classes of Ising models polynomial time algorithms are known which approximates $Z$ up to a $(1 \pm \varepsilon)$ approximation factor. In particular we use the following approximation algorithm for ferromagnetic[¶] Ising models due to Jerrum and Sinclair [JS93].

---

[¶]As pointed out by [Sri19], Jerrum and Sinclair's result (and hence, our result) extends to the *non-uniform external field* setting where there is a $\theta_i$ for each $i$ instead of $\theta_1 = \cdots = \theta_n = \theta$, with the restriction that each $\theta_i \geqslant 0$.

**Theorem C.2.** *There is an algorithm which given the parameters of a ferromagnetic Ising model distribution $P$, in $O(\varepsilon^{-2}n^{17}\log n)$ time returns a number $\hat{Z}$ such that with probability at least 9/10, $(1-\varepsilon)Z \leqslant \hat{Z} \leqslant (1+\varepsilon)Z$, where $Z$ is the partition function of $P$.*

Combining the previous two results with our general distance estimation algorithm, we can now obtain our main result for Ising models which we restate below.

**Theorem 4.2.** *Let $\mathcal{D}$ be the family of ferromagnetic Ising models having width at most $d$. Then, there is a distance approximation algorithm for $\mathcal{D}$ with sample complexity $m = e^{O(d)}\varepsilon^{-4}n^8\log(\frac{n}{\varepsilon})$ and runtime $O(mn^2 + \varepsilon^{-2}n^{17}\log n)$.*

*Proof.* We first use Theorem C.1 to get the parameters for a pair of Ising models $\hat{P}$ and $\hat{Q}$ which are, with probability at least $9/10$, pointwise $(1 \pm \varepsilon/8)$ approximations to $P$ and $Q$. If $\hat{P}$ or $\hat{Q}$ has any negative pairwise interaction term, then we modify them to zero, thus making $\hat{P}$ and $\hat{Q}$ ferromagnetic. We claim that since $P$ and $Q$ are ferromagnetic to start with, this can only improve the approximation factor. The reason is that Klivans and Meka, in their proof of Theorem C.1, show the more general result that for any *log-polynomial distribution*, i.e, any distribution $P$ on $\{-1,1\}^n$ where $P(x) \propto \exp(T(x))$ for a bounded-degree polynomial $T$, they can obtain a polynomial $\hat{T}$ with the same degree that satisfies a bound on $\|T - \hat{T}\|_1 = \sum_\alpha |T[\alpha] - \hat{T}[\alpha]|$ where $T[\alpha]$ and $\hat{T}[\alpha]$ are the coefficients of the monomial indexed by $\alpha$. It is clear that if $T[\alpha] \geqslant 0$, changing $\hat{T}[\alpha]$ to $\max(0, \hat{T}[\alpha])$ can only reduce $\|T - \hat{T}\|_1$.

Abusing notation for simplicity, henceforth let $\hat{P}$ and $\hat{Q}$ be the distributions after this modification. Let $N_{\hat{P}}(x)$ and $N_{\hat{Q}}(x)$ be the numerators for $\hat{P}$ and $\hat{Q}$ respectively. Then we apply Theorem C.2 to estimate, with probability $4/5$, the partition functions $\hat{Z}_P$ and $\hat{Z}_Q$ of $\hat{P}$ and $\hat{Q}$ respectively up to a $(1 \pm \varepsilon/8)$ multiplicative factor. Therefore, $E_P(x) = N_{\hat{P}}(x)/\hat{Z}_P$ and $E_Q(x) = N_{\hat{Q}}(x)/\hat{Z}_Q$ are $(\varepsilon/8, \varepsilon/4)$-EVAL approximators for $P$ and $Q$ respectively, where the $\varepsilon/8$-close distributions are $\hat{P}$ and $\hat{Q}$. It follows from Theorem A.1 that conditioned on the above, we can estimate $d_{\mathrm{TV}}(P,Q)$ up to an $\varepsilon$ additive error with probability at least $9/10$. □

**Remark C.3.** Klivans and Meka [KM17] have also given an algorithm for recovering the underlying dependency graph of an $n$-dimensional ising model using $O(\exp(O(d)/\eta^4)\log(\frac{n}{\eta\rho}))$ samples assuming its width at most $d$ and $\min_{i,j:A_{i,j} \neq 0}|A_{i,j}| \geqslant \eta$. Devroye et al. [DMR+20] have given a minimax-optimal algorithm that given the underlying graph, learns an Ising model in $d_{\mathrm{TV}} \leqslant \varepsilon$ using $O(n^2/\varepsilon^2)$ samples with 9/10 probability. These two results can be daisy-chained to improve the sample complexity of learning an unknown ising model and hence of our distance approximation algorithm. However, as noted in Section 6 of the later paper, this algorithm is not polynomial time and hence we will not get a polynomial time algorithm for distance approximation.

## C.1 Distance to uniformity

Next we give an algorithm for estimating the distance between an unknown Ising model and the uniform distribution over $\{-1,1\}^n$. We use the following recent result by Narayanan [Nar21].

**Theorem C.4** (Restated from [Nar21])**.** *Let $U$ and $D$ be the uniform distribution and any other distribution over $[N]$ respectively, such that we can sample from $D$, as well as compute the ratio $D(i)/D(j)$ for any $i \neq j \in [N]$ up to $(1 \pm \varepsilon)$ error for any $0 < \varepsilon < 1$ in unit time. Then $d_{\mathrm{TV}}(D,U)$ can be approximated up to an additive error using $\widetilde{O}(\varepsilon^{-2})$ samples with 2/3 probability.*

**Theorem 4.3.** *There is an algorithm which, given independent samples from an unknown Ising model $P$ over $\{-1,1\}^n$ with width at most $d$, takes $m = O(e^{O(d)}\varepsilon^{-4}n^8\log(n/\varepsilon))$ samples, $O(mn^2)$ time and returns a value $e$ such that $|e - d_{\mathrm{TV}}(P,U)| \leqslant \varepsilon$ with probability at least 7/12, where $U$ is the uniform distribution over $\{-1,1\}^n$.*

*Proof.* We first learn the parameters of the unknown ising model from samples using Theorem C.1. As we noted earlier computing the partition function naively is intractable in general. However computing $N_x/N_z$, the ratio of the probabilities of two items $x, y$ can be computed in $O(n^2)$ time up to $(1 \pm \varepsilon)$ approximation from Theorem C.1. The result follows from Theorem C.4. □

## D Multivariate Gaussians

In this section we give an algorithm for additively estimating the total variation distance between two unknown multidimensional Gaussian distributions. For a mean vector $\mu \in \mathbb{R}^n$ and a positive definite covariance matrix $\Sigma \in \mathbb{R}^{n \times n}$, the Gaussian distribution $N(\mu, \Sigma)$ has the pdf:

$$N(\mu, \Sigma; x) = \frac{1}{(2\pi)^{n/2}\sqrt{\det(\Sigma)}} \exp\left(-\frac{1}{2}(x-\mu)^{\top}\Sigma^{-1}(x-\mu)\right) \tag{5}$$

We use the following folklore result (see [ABDH$^+$17] for a proof) for learning the two Gaussians.

**Theorem D.1.** *Let $P$ be an $n$-dimensional Gaussian distribution. Let $\hat{\mu} \in \mathbb{R}^n$ and $\hat{\Sigma} \in \mathbb{R}^{n \times n}$ be the empirical mean and the empirical covariance defined by $O(n^2\varepsilon^{-2})$ samples from $P$. Then, with probability at least $9/10$, the distribution $\hat{P} = N(\hat{\mu}, \hat{\Sigma})$ satisfies $d_{\mathrm{TV}}(P, \hat{P}) \leqslant \varepsilon$.*

We are now ready to prove Theorem 4.4 restated below.

**Theorem 4.4.** *Let $\mathcal{D}$ be the family of multivariate gaussian distributions, $\{N(\mu, \Sigma) : \mu \in \mathbb{R}^n, \Sigma \in \mathbb{R}^{n \times n}, \Sigma \succ 0\}$. Then, there is a distance approximation algorithm for $\mathcal{D}$ with sample complexity $O(n^2\varepsilon^{-2})$ and runtime $O(n^\omega \varepsilon^{-2})$ (where $\omega > 2$ is the matrix multiplication constant).*

*Proof.* We first apply Theorem D.1 to obtain $\hat{P}$ and $\hat{Q}$ such that each is within $\varepsilon/4$ distance from $P$ and $Q$ respectively. Since we can evaluate the pdf of $\hat{P}$ and $\hat{Q}$ exactly, they serve as $(\varepsilon/4, 0)$ EVAL -approximators for $P$ and $Q$. Each determinant computation costs $O(n^\omega)$ time. Subsequently from (the continuous analog of) Theorem A.1, using $O(\varepsilon^{-2})$ samples from $P$ and $O(n^\omega \varepsilon^{-2})$ time, we can estimate $d_{\mathrm{TV}}(P, Q)$ up to an additive $\varepsilon$ error with probability at least $4/5$. □

**Remark D.2.** The above time analysis uses the unrealistic real RAM model in which real number computations can be carried out exactly upto infinite precision. However, there are strongly polynomial time algorithms for computing matrix determinant and inverse [Gác18, Wil65], so that even in the more realistic word RAM model, the above algorithm runs in polynomial time.

## E Causal Bayesian Networks under Atomic Interventions

We describe Pearl's notion of causality from [Pea09]. Central to his formalism is the notion of an *intervention*. Given a variable set $V$ and a subset $S \subset V$, an intervention $\mathrm{do}(s)$ is the process of fixing the set of variables in $S$ to the values $s$. If the original distribution on $V$ is $P$, we denote the *interventional distribution* as $P_s$, intuitively, the distribution induced on $V$ when an external force sets the variables in $S$ to $s$.

Another important component of Pearl's formalism is that some variables may be hidden (latent). The hidden variables can neither be observed nor be intervened upon. Let $V$ and $U$ denote the subsets corresponding to observable and hidden variables respectively. Given a directed acyclic graph $H$ on $V \cup U$ and a subset $S \subseteq (V \cup U)$, we use $\Pi_H(S)$ and $\mathrm{Pa}_H(S)$ to denote the set of all parents and observable parents respectively of $S$, excluding $S$, in $H$. When the graph $H$ is clear, we may omit the subscript.

**Definition E.1** (Causal Bayesian Network). *A (semi-Markovian) causal Bayesian network (CBN) on variables $X_1, \ldots, X_n$ is a collection of interventional distributions defined by a tuple $\langle V, U, G, \{\mathbf{Pr}[X_i \mid x_{\Pi(i)}] : i \in V, x_{\Pi(i)} \in \Sigma^{|\Pi(i)|}\}, \mathbf{Pr}[X_U]\rangle$, where (i) $G$ is a directed acyclic graph on $V \cup U = [n]$, (ii) $\mathbf{Pr}[X_i \mid x_{\Pi(i)}]$ is the conditional probability distribution of $X_i$ given that its parents $X_{\Pi(i)}$ take the values $x_{\Pi(i)}$, and (iii) $\mathbf{Pr}[X_U]$ is the distribution of the hidden variables $\{X_i : i \in U\}$.*

*A CBN $\mathcal{P} = \langle V, U, G, \{\mathbf{Pr}[X_i \mid x_{\Pi(i)}] : i \in V, x_{\Pi(i)} \in \Sigma^{|\Pi(i)|}\}, \mathbf{Pr}[X_U]\rangle$ defines a unique interventional distribution $P_s$ for every subset $S \subseteq V$ (including $S = \emptyset$) and assignment $s \in \Sigma^{|S|}$, as follows. For all $x \in \Sigma^{|V|}$:*

$$P_s(x) = \begin{cases} \sum_u \prod_{i \in V \setminus S} \mathbf{Pr}[x_i \mid x_{\pi(i)}] \cdot \mathbf{Pr}[X_U = u] & \textit{if } x \textit{ is consistent with } s \\ 0 & \textit{otherwise.} \end{cases}$$

Figure 1: An acyclic directed mixed graph (ADMG) where the bidirected edges are depicted as dashed. The in-degree of the graph is 2. The c-components are $\{A, C\}$ and $\{B, D, E\}$.

*We use $P$ to denote the observational distribution ($S = \emptyset$). $G$ is said to be the* causal graph *corresponding to the* CBN $\mathcal{P}$.

It is standard in the causality literature [TP02b, VP90, ABDK18] to assume that each variable in $U$ is a source node with exactly two children from $V$, since there is a known algorithm [TP02b, VP90] which converts a general causal graph into such graphs. Given such a causal graph, we remove every source node $Z$ from $G$ and put a *bidirected* edge between its two observable children $X_1$ and $X_2$. We end up with an Acyclic Directed Mixed Graph (ADMG) graph $G$, having vertex set $V$ and having edge set $E^{\rightarrow} \cup E^{\leftrightarrow}$ where $E^{\rightarrow}$ are the directed edges and $E^{\leftrightarrow}$ are the bidirected edges. The *in-degree* of $G$ is the maximum number of directed edges coming into any vertex in $V$. A *c-component* refers to any maximal subset of $V$ which is interconnected by bidirected edges. Then $V$ gets partitioned into c-components: $S_1, S_2, \ldots, S_\ell$. Figure 1 shows an example.

Throughout this section, we focus on *atomic* interventions, i.e. interventions on a single variable. Let $A \in V$ correspond to this variable. Without loss of generality, suppose $A \in S_1$. Tian and Pearl [TP02a] showed that in an ADMG $G$ as above, $P_a$ can be completely determined from $P$ for all $a \in \Sigma$ iff the following condition holds.

**Assumption E.2** (Identifiability wrt $A$). *There does not exist a path of bidirected edges between $A$ and any child of $A$. Equivalently, no child of $A$ belongs to $S_1$.*

Recently algorithms and sample complexity bounds for learning and sampling from identifiable atomic interventional distributions were given in [BGK$^+$20] under the following additional assumption. For $S \subseteq V$, let $\mathsf{Pa}^+(S) = S \cup \mathsf{Pa}(S)$.

**Assumption E.3** ($\alpha$-strong positivity wrt $A$). *Suppose $A$ lies in the c-component $S_1$, and let $Z = \mathsf{Pa}^+(S_1)$. For every assignment $z$ to $Z$, $P(Z = z) > \alpha$.*

We state the two main results of [BGK$^+$20], which given sampling access to the observational distribution $P$ of an unknown causal Bayesian network on a known ADMG return an $(\varepsilon, 0)$-EVAL approximator and an approximate generator for $P_a$. For the two results below, suppose the CBN $\mathcal{P}$ satisfies identifiablity (Assumption E.2) and $\alpha$-strong positivity (Assumption E.3) with respect to a variable $A \in V$. Let $d$ denote the maximum in-degree of the graph $G$ and $k$ denote the size of its largest c-component.

**Theorem E.4** (EVAL approximator and Sampler [BGK$^+$20]). *For any intervention $a$ to $A$ and parameter $\varepsilon \in (0, 1)$, there is an algorithm that takes $m = \tilde{O}\left(\frac{|\Sigma|^{2kd}n}{\alpha^k \varepsilon^2} \log \frac{1}{\delta}\right)$ samples from $P$, and in $O(mn \log^2 \frac{1}{\delta})$ time, returns a distribution $\hat{P}_a$ such that $d_{\mathrm{TV}}(P_a, \hat{P}_a) \leqslant \varepsilon$ with probability at least $(1 - \delta)$ and returns a circuit $E_{P,a}$ such that:*

   – *Evaluation: Given an assignment $x$ to the nodes, $E_{P,a}$ outputs $\hat{P}_a(x)$ exactly in $O(n)$ time.*

   – *Generation: Obtaiing an independent sample from $\hat{P}_a$ takes $O(n)$ time.*

We give a distance approximation algorithm for identifiable atomic interventional distributions using the above result and Theorem A.1.

**Theorem E.5** (Formal version of Theorem 4.6). *Suppose $\mathcal{P}, \mathcal{Q}$ are two unknown CBN's on two known ADMGs $G_1$ and $G_2$ on a common observable set $V$ both satisfying Assumption E.2 and*

*Assumption E.3 wrt a special vertex A. Let $d$ denote the maximum in-degree, and $k$ denote the size of the largest c-component of $G_1$ and $G_2$.*

*Then there is an algorithm which for any $a \in \Sigma$ and parameter $\varepsilon \in (0,1)$, takes $m = \tilde{O}\left(\frac{|\Sigma|^{2kd}n}{\alpha^k \varepsilon^2} \log \frac{1}{\delta}\right)$ samples from $P$ and $Q$, runs in time $\tilde{O}(mn \log^2 \frac{1}{\delta})$ and returns a value $e$ such that $|e - d_{\mathrm{TV}}(P_a, Q_a)| \leqslant \varepsilon$ with probability at least 2/3.*

*Proof.* We first invoke Theorem E.4 to learn the two distributions as $\hat{P}_a$ and $\hat{Q}_a$ with the distance parameter $\varepsilon$, which serve as $(\varepsilon, 0)$-EVAL approximators for $P_a$ and $Q_a$ respectively. Once learnt, no further samples are needed from $P_a$ and $Q_a$. $\hat{P}_a$ can be sampled in $O(n)$ time from Theorem E.4. The result follows from Theorem A.1. $\qquad\square$

## F  Improving Success of Learning Algorithms Using Distance Estimation

In this section we give a general algorithm for improving the success probability of learning certain families of distributions. Specifically, let $\mathcal{D}$ be a family of distributions for which we have a learning algorithm $\mathcal{A}$ in $d_{\mathrm{TV}}$ distance $\varepsilon$ that succeeds with probability 3/4. Suppose there is also a distance approximation algorithm $\mathcal{B}$ for $\mathcal{D}$. The algorithm presented below, which uses $\mathcal{A}$ and $\mathcal{B}$, learns an unknown distribution from $\mathcal{D}$ with probability at least $(1 - \delta)$.

---

**Algorithm 4:** High probability distribution learning

    **Data:** Samples from an unknown distribution $P$
    **Result:** A distribution $\hat{P}$ such that $d_{\mathrm{TV}}(P, \hat{P}) \leqslant \varepsilon$ with probability $1 - \delta$
**1** **for** $0 \leqslant i \leqslant R = O(\log \frac{1}{\delta})$ **do**
**2**     $P_i \leftarrow$ Run $\mathcal{A}$ on samples from $P$ to get a learnt distribution;
**3**     $count_i \leftarrow 0$;
**4** **for** *every unordered pair* $0 \leqslant i < j \leqslant R$ **do**
**5**     $d_{ij} \leftarrow$ Estimate distance between $P_i$ and $P_j$ up to additive error $\varepsilon$ using $\mathcal{B}$;
**6**     **if** $d_{ij} \leqslant 3\varepsilon$ **then**
**7**        $count_i \leftarrow count_i + 1$;
**8**        $count_j \leftarrow count_j + 1$;
**9** $i^* = \arg\max_i count_i$;
**10** **return** $P_{i^*}$;

---

**Theorem F.1.** *Let $\mathcal{D}$ be a family of distributions. Suppose there is a learning algorithm $\mathcal{A}$ which for any $P \in \mathcal{D}$ takes $m_{\mathcal{A}}(\varepsilon)$ samples from $P$ and in time $t_{\mathcal{A}}(\varepsilon)$ outputs a distribution $P_1$ such that $d_{\mathrm{TV}}(P, P_1) \leqslant \varepsilon$ with probability at least 3/4. Suppose there is a distance approximation algorithm $\mathcal{B}$ for $\mathcal{D}$ that given any two completely specified distributions $P_1$ and $P_2$ estimates $d_{\mathrm{TV}}(P_1, P_2)$ up to an additive error $\varepsilon$ in $t_{\mathcal{B}}(\varepsilon, \delta)$ time with probability at least $(1 - \delta)$. Then there is an algorithm that uses $\mathcal{A}$ and $\mathcal{B}$ as subroutines, takes $O(m_{\mathcal{A}}(\varepsilon/4) \log \frac{1}{\delta})$ samples from $P$, runs in $O(t_{\mathcal{A}}(\varepsilon/4) \log \frac{1}{\delta} + t_{\mathcal{B}}(\varepsilon/4, \frac{\delta}{210000 \log^2 \frac{2}{\delta}}) \log^2 \frac{1}{\delta})$ time and returns a distribution $\hat{P}$ such that $d_{\mathrm{TV}}(P, \hat{P}) \leqslant \varepsilon$ with probability at least $1 - \delta$.*

*Proof.* The boosting algorithm is given in Algorithm 4. We take $R = 324 \log \frac{2}{\delta}$ repetitions of $\mathcal{A}$ to get the distributions $P_i$s. From Chernoff's bound at least $2R/3$ distributions (successful) satisfy $d_{\mathrm{TV}}(P_i, P) \leqslant \varepsilon$ with probability at least $1 - \delta/2$, which we condition on henceforth. These successful distributions have pairwise distance at most $2\varepsilon$. Conditioned on the $\binom{R}{2}$ calls to $\mathcal{B}$ succeeding, the pairwise distances between the successful distributions are at most $3\varepsilon$. Hence every successful $i$ has its count value at least $2R/3 - 1$. This means $i^*$, which has the maximum count value ($\geqslant 2R/3 - 1$) must intersect at least one successful $i'$ such that $d_{\mathrm{TV}}(P_{i^*}, P_{i'}) \leqslant 3\varepsilon$. By triangle inequality we get $d_{\mathrm{TV}}(P_{i^*}, P) \leqslant 4\varepsilon$.

It suffices for each call to $\mathcal{B}$ succeed with probability at least $\frac{\delta}{2R^2}$. $\qquad\square$

Assuming black-box access to $\mathcal{A}$, $O(m_{\mathcal{A}} \log \frac{1}{\delta})$ samples are needed in the worst case to learn with $1 - \delta$ probability, since otherwise all the $o(\log \frac{1}{\delta})$ repetitions may fail. We can apply the above

algorithm to improve the success probability of learning Bayesian networks on a given graph with small indegree.