[Reviews · NeurIPS 2020]

Review 1

Summary and Contributions: * Extends the framework / algorithm presented in [CR14] with the idea of EVAL approximators. Gives a new algorithm for approximating the distance between two probability distributions and gives theoretical guarantees for sample complexity. * Applies this framework to four settings: Bayesian networks, Ising models, multivariate Gaussians, causal models, extending or improving upon existing results.

Strengths: The central idea is conceptually simple, which makes it adaptable to a variety of settings. The claims in this paper are supported by theoretical proofs. The topic addressed (distance learning / testing) is highly applicable to the broader Statistics / ML community.

Weaknesses: The novelty of this paper over [CR14], upon which the central result of this paper is based, is not immediately clear to me. However, this may simply be because I am not as familiar with the surrounding research. I discuss this in greater detail in the "Relation to prior work" section of my review. The paper would benefit from a more thorough discussion of the differences between the two papers, and how Algorithm 1 enables results that were not possible using the ideas presented in [CR14]. The paper would also benefit from empirical studies of its performance in practice.

Correctness: The central results of the paper are supported by proofs in the Appendix. No empirical / simulation studies are given.

Clarity: The ideas are presented clearly. The paper overall could benefit from restructuring, in my opinion: for example, moving Section 2.6 (Previous work) into or just following the introduction; giving the main conceptual result its own section; and putting in another sections the implications of the conceptual result for the various illustrative models.

Relation to Prior Work: The main conceptual contribution (Algorithm 1, the use of EVAL approximators) is a direct extension of [CR14], as mentioned by the authors in Section 2.1 and at the end of Section 2.6. This paper would benefit from a more thorough discussion of of the limitations of [CR14] in contrast to the paper under consideration. In particular, it would be useful to understand why the results that flow from Algorithm 1 would not have been directly attainable through the framework proposed in [CR14]. In terms of the results that flow from Algorithm 1, the authors lay out clearly the limitations of past work in testing for the models considered (Bayesian etworks, Ising models, multivariate Gaussians, etc.)

Reproducibility: Yes

Additional Feedback: *** UPDATE AFTER REBUTTAL *** I thank the authors for addressing my concerns regarding the novelty of this paper, especially over [CR14]. I am glad to hear that the authors intend to include their extended comparison with [CR14] in the paper itself (ideally, as part of Section 2.6). I adjusted my overall score accordingly. Typo that I spotted: line 61 - should "class of" be "classes of"?


Review 2

Summary and Contributions: This paper considers estimating the statistical distance between two generative models given sample access from the two models. The general framework proceeds by first learning the model, and then a simple unbiased estimator of the statistical distance can be applied with the learnt model for distance estimation. The framework is applied to bayesian networks, Ising models, gaussian distributions and causal models. In particular, a new learning algorithm for bayesian networks on a known DAG G is introduced.

Strengths: First computational and sample efficient distance approximation algorithm for a variety of structured high dimensional distributions.

Weaknesses: The approach is rather straightforward, and the technical novelty is not clear.

Correctness: Yes

Clarity: Yes

Relation to Prior Work: Yes

Reproducibility: Yes

Additional Feedback: Post rebuttal comment: After discussion with other reviewers, I think the paper is somewhat between 6 and 7. Therefore I choose not change my score.


Review 3

Summary and Contributions: The paper is concerned with the problem of total variation distance estimation in a variety of prominent high dimensional structured distributions. The principle contribution is a scheme centered around the use of EVAL approximators. A (beta, gamma) EVAL approximator for a law P is a function E_P with an associated distribution \hat{P} such that TV(P, \hat{P}) \le \beta, and for any x, |E_P(x)/P(x) - 1| \le \gamma. (0,0) EVAL queries were first used in CR14 to develop strong testing/distance approximation schema. Thm 2.3 of this paper extends this is (\epsilon, \epsilon) EVAL queries, showing that O(\epsilon^{-2}) samples from a distribution P and O(\epsilon^{-2}) queries to such EVAL approximators for two laws P and Q allows efficient estimation of TV(P,Q) to error O(\epsilon). This allows the authors to construct efficient distance approximation (and tolerant testing) schema by developing EVAL approximators. This is pursued in a variety of settings by utilising recently developed methods in the literature that learn efficient approximations to distributions in the relevant classes.

Strengths: I find the method underlying Thm 2.3 to be clever, simple, and flexible. The resulting bounds on sample/time complexity of distance approximation and tolerant testing are novel, and are developed for high dimensional families relevant to the recent literature.

Weaknesses: To me the main weakness of the paper is insufficient discussion of how tight these results are. I'll focus on sample complexity. The main discussion regarding this (starting line 141) argues that since for completely unstructured distributions and for product distributions on binary cubes the sample complexities of learning are not too separated from that of distance approximation. However, to me these are natural edge cases of high dimensional models, and I don't find it implausible that something different can occur in the bulk of the problem. This matters because of features such as the exponential dependence on width in the case of Ising models that occur in the derived bounds, which don't appear in non-tolerant testing. (Of course, in this case, this could be due to the fact that the learnt \hat{P} in KM has |\hat{P}(x)/P(x) - 1| \le \epsilon for every x, which is much stronger than TV(\hat{P}, P) < \epsilon.) I think clear discussion of if the individual bounds of secs 2.2-2.5 are (expected to be) tight or not would go a far way in strengthening this aspect of the paper.

Correctness: Yes.

Clarity: I found the paper well written and easy to read. The problem is contextualised well, and the main idea is well explained. I also appreciate the effort taken to ground the discussion of each of the models studied.

Relation to Prior Work: Yes. To the best of my knowledge, most relevant work is discussed, and the differences well delineated.

Reproducibility: Yes

Additional Feedback: --- Section 2.1 and 2.2-2.5 and 2.6 seem to me to be thematically separate sections. The first sets up the underlying method, and context, the next four can be viewed as applications of this when coupled with learning methods, and the final is clearly separate. Reorganising along these lines should smoothen the presentation. --- I think the argument of Corollary D.3 is simple and short enough to be pushed to the main text. --- for Ising models - [1] describes the minimax rate of learning an Ising model in total variation given the underlying graph of the model. In KM17, the sample complexity of estimating the graph is much smaller than that of approximation in the sense used in Thm C.1 (there's no n^8). Can these be daisy chained to improve the sample costs of Thm. 2.5? [1]: arxiv.org/abs/1806.06887

[Author Response · NeurIPS 2020]

We'd like to sincerely thank all the reviewers for a careful reading of our paper in these difficult times and welcome their suggestions. Below, we respond to each reviewer individually.

**Response to Reviewer #1:**

- [Novelty of our work] We would like to emphasize that the core conceptual novel contribution of our work is the establishment of connection between the testing in the dual access model (and in the conditional sampling model) to testing and distance approximation in the standard sampling model. These two models have been investigated separately. Here we use the former results to derive several new efficient tolerant testing algorithms in the standard model for high dimensional distributions, thus extending the state-of-the-art in this area. In this regard, we extend [CR14] to derive Algorithm 1, which in our view is intended to be simple and flexible, as acknowledged by Reviewer 4 as well. We consider the simplicity of Algorithm 1 a core strength of our work.

- [Comparison with [CR14]] Technically, [CR14] assumes perfect access to the probability mass functions of the two distributions. Instead we work with approximate access to p.m.f.s, the approximation being parameterized by $\beta$ and $\gamma$. In our opinion, the generalization (in Appendix A.1) does not follow trivially. The usage of approximation has allowed us to obtain results for several high dimensional distributions that do not follow from [CR14]. For example, let us consider the Ising model. In this case, given samples from two ferromagnetic Ising models $P$ and $Q$, we approximately learn the model parameters [KM17] and estimate the partition functions [JS93], to evaluate the p.m.f.s approximately. The later result takes parameters of a ferromagnetic Ising model as input and returns a (randomized) PTIME $(1 \pm \epsilon)$-multiplicative approximation of its partition function, and therefore we obtain a PTIME algorithm. In contrast, since the computation of the partition function given a fully known ferromagnetic Ising model is known to be #P-complete [JS93] (Theorem 15) and as [CR14] does not allow for multiplicative errors, [CR14] would lead to an algorithm with $\mathsf{P}^{\#\mathsf{P}}$ complexity. As R1 pointed out, we should have included the above discussion in the paper itself. We will do so in the final version contrasting the limitations of directly using [CR14] for all the different families.

We appreciate and agree with your valuable thoughts regarding the restructuring of the paper. We will incorporate them in the final version to the extent possible.

**Response to Reviewer #3:**

We appreciate this reviewer for acknowledging our Bayes net learning algorithm from the Appendix. Please see our response to the Reviewer #1 presented above for the other novelties of our work.

**Response to Reviewer #4:**

- [Tightness of our bounds] We note that the $\Omega(n/\log n)$ lower bound from [CDKS17] that we cite on line 152-154 for tolerant testing of production distributions implies the same lower bound for tolerant testing of Bayes nets. Same lower bound also holds for tolerant testing for the class of atomic interventional distributions. For the Ising model, currently we do not have a lower bound and in general, we agree with the reviewer that proving improved lower bounds for the tolerant testing problems we consider here is an important open direction - in this paper our focus was mainly establishing upper bounds. We will include this discussion in the final version.

- [Using a better learning algorithm for Ising models] Thanks a lot for bringing arXiv:1806.06887 to our notice. The approach you suggested should give a substantially better sample complexity dependence on $n$, for learning the Ising model. Since we plug in the learning algorithm as a black-box, this directly improves sample complexity of our algorithm. However, as noted in Section 6 of their paper, this algorithm is not polynomial time and hence we will not get a polynomial time algorithm.

We appreciate and agree with the rest of your suggestions for improving the presentation of the paper and we will incorporate them in the final version as appropriate.

[Meta-Review · NeurIPS 2020]

While the technical delta from prior works in the paper is not very big, there are nice conceptual insights that should of broader interest to the community and generic enough so that they may be applicable to future work.